# Enhancement of the Antitumor and Antimetastatic Effect of Topotecan and Normalization of Blood Counts in Mice with Lewis Carcinoma by Tdp1 Inhibitors—New Usnic Acid Derivatives

**DOI:** 10.3390/ijms25021210

**Published:** 2024-01-19

**Authors:** Tatyana E. Kornienko, Arina A. Chepanova, Alexandra L. Zakharenko, Aleksandr S. Filimonov, Olga A. Luzina, Nadezhda S. Dyrkheeva, Valeriy P. Nikolin, Nelly A. Popova, Nariman F. Salakhutdinov, Olga I. Lavrik

**Affiliations:** 1Novosibirsk Institute of Chemical Biology and Fundamental Medicine, Siberian Branch of the Russian Academy of Sciences, 8, Akademika Lavrentieva Ave., Novosibirsk 630090, Russia; t.kornienko1995@gmail.com (T.E.K.); arinachepanova@mail.ru (A.A.C.); elpida80@mail.ru (N.S.D.); lavrik@niboch.nsc.ru (O.I.L.); 2N. N. Vorozhtsov Novosibirsk Institute of Organic Chemistry, Siberian Branch of the Russian Academy of Sciences, 9, Akademika Lavrentieva Ave., Novosibirsk 630090, Russia; alfil@nioch.nsc.ru (A.S.F.); luzina@nioch.nsc.ru (O.A.L.); anvar@nioch.nsc.ru (N.F.S.); 3Institute of Cytology and Genetics, Siberian Branch of the Russian Academy of Sciences, 10, Akademika Lavrentieva Ave., Novosibirsk 630090, Russia; nikolin@bionet.nsc.ru (V.P.N.); nelly@bionet.nsc.ru (N.A.P.)

**Keywords:** usnic acid derivatives, TDP1 inhibitors, anticancer therapy, topotecan

## Abstract

Tyrosyl-DNA phosphodiesterase 1 (Tdp1) is an important DNA repair enzyme and one of the causes of tumor resistance to topoisomerase 1 inhibitors such as topotecan. Inhibitors of this Tdp1 in combination with topotecan may improve the effectiveness of therapy. In this work, we synthesized usnic acid derivatives, which are hybrids of its known derivatives: tumor sensitizers to topotecan. New compounds inhibit Tdp1 in the micromolar and submicromolar concentration range; some of them enhance the effect of topotecan on the metabolic activity of cells of various lines according to the MTT test. One of the new compounds (compound **7**) not only sensitizes Krebs-2 and Lewis carcinomas of mice to the action of topotecan, but also normalizes the state of the peripheral blood of mice, which is disturbed in the presence of a tumor. Thus, the synthesized substances may be the prototype of a new class of additional therapy for cancer.

## 1. Introduction

The development of inhibitors of important enzymes and DNA repair factors is one of the promising areas of modern pharmacology and is one of the ways to create effective cancer therapy. Such compounds should be able to inhibit key DNA repair enzymes, which would significantly enhance the effectiveness of traditional treatments. 

Tyrosyl DNA phosphodiesterase 1 (Tdp1) is a promising target for cancer treatment [1,2,3,4] because it plays a key role in removing DNA damage formed by camptothecin group compounds (irinotecan and topotecan), widely used in anticancer therapy [5,6]. In addition, Tdp1 is involved in the removal of DNA damages caused by other anticancer drugs common in clinical practice (temozolomide, bleomycin, etoposide, etc.) [1]. The mechanisms of action of the listed drugs are different, as well as the repair ensembles of proteins that remove damage [7,8,9,10,11,12]. The hypothesis that Tdp1 is responsible for the drug resistance of some types of cancer [13] is supported by a number of studies: Tdp1 knockout mice and human cell lines that have a mutation that reduces the activity of this enzyme are hypersensitive to camptothecins [8,14,15,16,17]. In addition, suppression of Tdp1 expression with minocycline enhances the antimetastatic effect of irinotecan and increases the lifespan of experimental animals [18]. Conversely, in cells with increased expression levels, camptothecin and etoposide cause less DNA damage [9,19]. Also, the response of tumors overexpressing Tdp1 to irinotecan is reduced [20]. The above data suggest that the combination of anticancer drugs and Tdp1 inhibitors can significantly improve the effectiveness of chemotherapy. 

The best-studied substrate for Tdp1, from which it receives its name, is the covalent adduct of tyrosine 723 (in humans) of topoisomerase 1 (Top1) with the 3-terminus of DNA, a short-lived intermediate of the normal catalytic cycle of Top1 [21]. This intermediate can be stabilized by DNA damage that changes its geometry, or by camptothecin and its derivatives, Top1 inhibitors, which have a powerful antitumor effect [5,22,23]. We have focused our efforts on developing Tdp1 inhibitors that, in combination with topotecan, enhance the antitumor properties of the latter.

Over the past few decades, hundreds of Tdp1 inhibitors have been discovered, both among natural substances isolated from plants, fungi, and marine organisms, and among aminoglycoside antibiotics and steroid derivatives. Most studies of these substances have not progressed beyond purified enzyme inhibition and cytotoxic properties in passaged cell lines [24,25,26,27].

Our recently discovered Tdp1 inhibitors based on natural biologically active substances sensitize the antitumor effect of topotecan in vivo in mouse tumor models reviewed in [27]. All of them inhibit Tdp1 at submicromolar concentrations. The sensitizing properties of these compounds are manifested both on cultured cell lines, when the cytotoxic effect of topotecan is enhanced, and in enhancing the antitumor and antimetastatic effect of topotecan in vivo. 

Thus, Tdp1 inhibitors of various chemical nature are able to enhance the cytotoxic and antitumor effect of topotecan, which indicates their promise for the further development of accompanying therapy for oncological diseases.

Compounds **1** and **2**, which belong to the type of enamine usnic acid derivatives (Figure 1), are nontoxic to the different cell types [28,29] and are able to enhance the effect of topotecan in vitro against tumor cells and the effect of topotecan in vivo (only compound **1** was studied) [30]. Compound **1** was shown to be well tolerated and the antitumor effect of topotecan was enhanced at least twofold when exposed to mouse Lewis lung adenocarcinoma, as well as there being a significant increase in the antimetastatic effect, which obviously confirms the prospects and necessity for further research of this class of compounds as modern agents for accompanying anticancer therapy. 

Compounds **3** and **4** belong to the type of hydrazonothiazole derivatives of usnic acid (Figure 1). Compound **3** is moderately toxic to cells (CC_50_ values 9–16 µM) and enhances the cytotoxicity of topotecan [31]. Compound **4** is somewhat more toxic to cells (CC_50_ values of 1.5 μM), but its sensitizing activity and good tolerability were confirmed in experiments both on MCF-7 cells and in experiments on mice with Lewis lung carcinoma [32].

In this work, we proposed to synthesize a compound based on usnic acid that combines both pharmacophore fragments in its structure. We hypothesize that modification of the hydrazinothiazole compound to form an enamine may reduce the toxicity of the compound while maintaining their inhibitory properties. It is known that the OH-3 group is responsible for the protonophore activity of usnic acid, which in turn is responsible for the uncoupling of oxidative phosphorylation and determines the toxicity of usnic acid [33]. When the OH-3 group is modified, the protonophore activity of usnic acid is significantly reduced [34]. Thus, modification of the triketone fragment of this molecule can reduce the protonophore activity of the resulting compounds and lead to a decrease in their toxicity. Combining enamine and hydrazonothiazole substituents, we obtain four new compounds (Figure 2).

Thus, we have developed methods for the synthesis of such hybrid compounds, studied their inhibitory and cytotoxic properties, as well as their ability to enhance the effect of topotecan in vitro. The new compounds have significantly less effect on the survival of all types of cells than the original compound **4**; some of them enhance the effect of topotecan on the metabolic activity of cells of various lines.

We further tested the ability of the lead compound **7** to potentiate the antitumor and antimetastatic effect of topotecan in a mouse model of Lewis lung carcinoma and the antitumor effect of topotecan on Krebs-2 carcinoma. We compared the effectiveness of different methods of drug administration (intraperitoneal, intragastric) and found that intraperitoneal administration of compound **7** in combination with topotecan was most effective in all cases. 

Next, we studied the acute toxicity of the compound alone and in combination with topotecan. In addition, we examined the effect of the compound on the peripheral blood of mice. Compound **7** does not cause acute toxicity and protects the peripheral blood condition impaired in Krebs-2 carcinoma.

## 2. Results and Discussion

### 2.1. Chemistry

The synthesis scheme for the hybrid derivatives **5**–**8** includes an initial modification of ring A and then ring C of usnic acid **9** (Figure 1). The synthesis of a compounds **5**–**8** was carried out by modification of ring A to the hydrazonothiazole derivatives **3**,**4**, followed by its introduction into a reaction with an amine (R_2_NH_2_, Figure 1). This modification was carried out starting from the derivatives with thiazole ring **3**,**4**, based on the assumption that the reverse order of assembly may be more inconvenient (primarily due to the ambiguity of bromination of usnic acid enamines).

The hydrazonothiazoles **3** and **4** were synthesized by a previously developed methodology (Figure 2) [31]. In the first step, bromination of usnic acid **9** with bromine in dioxane was carried out. The derivative **10** was isolated in 70% yield after column chromatography. Thiosemicarbazones of the corresponding aldehydes **11a**,**b** were also synthesized by reacting them with thiosemicarbazide in ethanol. In the second step, the reaction of bromousnic acid **11** with thiosemicarbazones **12** in methanol was carried out, yielding hydrazonothiazoles **3** and **4** in 75–78% yields.

For the synthesis of the target compounds, the starting hydrazinothiazole derivatives **3**,**4** were used. As the amine component, those amines were chosen for which modification of usnic acid led to an increase in its Tdp1 inhibitory properties and a decrease in toxicity: para-bromoaniline and 1-amino-3-(3,5-di-tert-butyl-4-hydroxyphenyl)propane. The reaction with amines was carried out in ethanol at boiling (Figure 3) according to methods previously developed [31,35,36,37,38,39]. The yields of target hybrid compounds **5**–**8** were 27–78%; purification was carried out by column chromatography. The yield of compounds with p-bromophenyl substituent (70 and 75%) was significantly higher than that of derivatives with 3-(2,6-ditretbutyl-3-hydroxyphenyl-4)-propyl substituent (27 and 34%), which may be related to their low stability under conditions of column chromatography.

Thus, new compounds containing enamine moieties in the C ring were obtained by modification of hydrazinothiazole derivatives **2**,**3** based on usnic acid **9**.

### 2.2. Biology

#### 2.2.1. Inhibitory Properties of Usnic Acid Derivatives against Tdp1

The inhibitory properties of the compound on the purified Tdp1 was studied using a technique developed previously by our team [27]. A 16-mer single-stranded oligonucleotide carrying a fluorophore at the 5′-end and a quencher at the 3′-end was used as a biosensor. When within the Förster radius, the quencher suppressed the fluorescence of the fluorophore. The quencher was removed by Tdp1, resulting in fluorescence emission. Data on Tdp1 inhibition are shown in Table 1. Usnic acid derivatives with a bromophenol residue in the enamine fragment (compounds **6** and **8**) inhibit the enzyme with half-maximal inhibition concentrations (IC_50_) an order of magnitude worse than derivatives with a di-tert-butyl fragment (compounds **5** and **7**).

#### 2.2.2. The Influence of Usnic Acid Derivatives on the Survival of Various Types of Cells

We further studied the cytotoxic/antiproliferative properties of the resulting compounds using the MTT test, since it is important that accompanying therapy based on Tdp1 inhibitors does not lead to an increase in already severe side effects. We found that compounds **5** and **7** with di-tert-butyl fragment have no or little effect on the metabolic activity of various types of cultured cells at concentrations up to 100 μM (more than 50% of living cells) against a wide panel of cell lines representing various tumor models, as well as cells of non-cancerous origin. Compounds **6** and **8** with a bromophenol residue in the enamine fragment have a weak or moderate effect on cell survival (Table 1 and Appendix A). 

We compared the effect of the “parent” hydrazonothiazole compound **4** on the metabolic activity of cells with the effect of new compounds using the MTT test. Compared to **4**, the new derivatives actually have a significantly less effect on cell survival according to MTT (Table 1), although their inhibitory properties are closer to the enamine derivatives of usnic acid.

#### 2.2.3. Sensitizing Properties of Usnic Acid Derivatives In Vitro

Currently available data indicate the fundamental ability of Tdp1 inhibitors to significantly enhance the therapeutic effect of topotecan [30,31,32,40,41,42]. Tdp1 inhibitors combined with topotecan dramatically reduce tumor metastasis [30,32]. Usnic acid derivatives of the enamine and hydrazonothiazole series turned out to be promising sensitizers of the action of topotecan both on cultured cell lines and on mouse tumors in vivo [30,32]. We studied the ability of the usnic acid derivatives to sensitize the effects of topotecan on the same panel of cell lines. We used two non-cancerous cell lines MRC-5 and HEK293A, including previously obtained HEK293A Tdp1 knockout cells [29] (for compound **7**), and cancerous cell lines HeLa, HCT-116, A-549, MCF-7, T98G. We found the most pronounced and reliable sensitizing effect on cancerous cell lines HeLa, HCT-116, A-549, MCF-7 only for compound **7** (Table 2 and Appendix A).

#### 2.2.4. Anticancer and Antimetastatic Effects of Compound **7** Used as Monotherapy or in Combination with Topotecan in the Lewis Lung Carcinoma Model

Since compound **7** was found to be the most effective and nontoxic Tdp1 inhibitor (Table 1), as well as a topotecan sensitizer for the maximum number of cell lines (Table 2), we selected this compound for further studies.

Compound **7** is a hybrid of two compounds: leaders in their subclasses of usnic acid derivatives **1** and **4** [28,32]. We studied the ability of compound **7** to sensitize the effect of topotecan on murine Lewis lung carcinoma (LLC) (Tpc + 7, Figure 3). The LLC cell line was maintained as a transplantable strain in mice and was provided by the cell depository of the Institute of Cytology and Genetics SB RAS (Novosibirsk, Russia). LLC is widely used as a model of lung cancer, and exhibits high tumorigenicity and metastasis to the lungs in C57BL mice [43,44,45,46]. Subcutaneous transplantation leads to the development of a primary node at the injection site and the appearance of metastases in the lungs on days 17–21 [47]. The size of the primary tumor was determined as the difference in the weight of the healthy limb and the limb with the tumor after the mice were removed from the experiment on 18th day after LLC transplantation. 

Figure 3A shows that the administration of topotecan (1 mg/kg intraperitoneally, i/p) reduced primary node size; the tumor growth inhibition (TGI) value was 36.4%, compared to intact control. The use of the combination of topotecan with 50 mg/kg compound **7** (intragastrical administration of compound **7**, i/g) leads to a further reduction in tumor weight, TGI was 47.7%. The administration of compound **7** i/g by itself did not lead to a decrease in tumor weight. The best results were obtained for the combination of topotecan with compound **7** (50 mg/kg, i/p), TGI was 57.2%; the difference was significant both in comparison with both controls and with the topotecan group (*p* < 0.05).

The effect of compound **7** on the antimetastatic properties of topotecan was less pronounced (Figure 3B). At the dose used, topotecan had virtually no effect on the number of metastases in the lungs, and the number of metastases was not affected by the use of compound **7** both i/p and i/g, as well as the combination of topotecan with compound **7** i/g.

The primary data and the results of their processing according to the Tukey criterion are given in the Appendix A.

The combination of topotecan with compound **7** i/p reduced the number of metastases, although the difference with the control and with the topotecan group was insignificant.

Thus, the use of compound **7** in mono mode has no effect on either the growth of the primary node or the number of metastases in the lungs. In combination with topotecan, there was a tendency to reduce the weight of the primary tumor and the number of metastases, which was more pronounced with intraperitoneal administration of compound **7**.

#### 2.2.5. Anticancer Effect of Compound 7 Used as Monotherapy or in Combination with Topotecan in the Krebs-2 Ascitic Carcinoma Model

The Krebs-2 cancer cell line was obtained from the cell depository of the Institute of Cytology and Genetics SB RAS (Novosibirsk, Russia), and is maintained in mice as a transplanted tumor. Krebs-2 carcinoma is nonspecific to the different genetic constitutions of mice and is weakly immunogenic for mice of all strains. The biological source of Krebs-2 is the epithelial cells of the abdominal wall of the mouse. With intraperitoneal transplantation of tumor cells, an ascitic form is formed [48,49,50,51,52]. 

The effect of the drugs was assessed by the weight of ascites and the number of tumor cells in it. 

The primary data and the results of their processing according to the Tukey criterion are given in Appendix A.

The combination of topotecan and compound **7** i/p was more effective in terms of ascites weight and the number of tumor cells in ascitic fluid than the same combination with compound **7** i/g. Also, the combination of topotecan and compound **7** i/p had a more pronounced antitumor effect than topotecan, compound **7** i/p, and compound **7** i/g separately (Figure 4).

Thus, the new compound **7** potentiates the antitumor and antimetastatic effect of topotecan when administered i/p not worse than “parent” hydrazonothiazole compound **4**, but we did not obtain the expected drastic enhancement of the effect compared to the “parent” compounds.

#### 2.2.6. The Toxic Effect of Drugs and Their Combination

We also studied the effect of topotecan, compound **7**, and their combinations on changes in body weight of mice with LLC from days 1 to 18 after tumor transplantation and of mice with Krebs-2 carcinoma from days 1 to 8. In both cases, a tendency towards an increase in the weight of animals was revealed, which indicates the absence of acute toxicity of the studied compounds (Figure 5).

In addition, at the end of the experiments, we removed and weighed the liver and spleen, and calculated the liver and spleen indices. No significant changes in the weight of liver and spleen were found (Appendix A). We also did not observe any changes in the behavior or appearance of the mice.

Thus, therapy with topotecan, compound **7**, and their combination was well tolerated and did not cause acute toxic effect.

#### 2.2.7. Effects of the Test Substances on the Peripheral Blood in Mice with Krebs-2 Carcinoma 

We also counted leukocytes and erythrocytes in mice with Krebs-2 carcinoma. As can be seen from the data in Table 3, mice with Krebs-2 carcinoma (intact control) have an increased number of leukocytes and a decreased number of erythrocytes compared to healthy mice. An increase in white blood cell count in the peripheral blood is associated with the inflammatory process that develops during tumor formation and growth. The administration of solvent for compound **7** (DMSO + Tween-80), topotecan alone, as well as topotecan in combination with compound **7** i/g, slightly reduced the number of leukocytes. It should be noted that topotecan at the dose of 1 mg/kg was not hemotoxic and partly normalized the concentrations of leukocytes and erythrocytes, apparently due to its antitumor effect. Treatment of mice with only compound **7**, regardless of the route of administration, reduced the number of leukocytes somewhat more effectively than the drugs listed above, but the decrease was significant only in comparison with the control without treatment. Administration of topotecan in combination with compound **7** i/p significantly reduced the number of leukocytes compared to the control without treatment, the control with DMSO + Tween-80 and the group of mice receiving only topotecan.

Mice with Krebs-2 carcinoma have a reduced number of red blood cells compared to healthy mice (Table 3), which may be caused by general intoxication of the body. The number of red blood cells practically did not change with the introduction of the solvent. At the same time, treatment with drugs and their combinations significantly increased the number of erythrocytes, regardless of the method of administration of compound **7**. The largest effect was again exerted by the combination of topotecan and compound **7** i/p, which increased the number of erythrocytes to healthy control.

Thus, the blood cell counts returned to normal values (red blood) or significantly improved (white blood) after treatment with combination Tpc + **7** i/p, i.e., hematopoiesis was normalized (Table 3). We used the values observed in untreated healthy mice without tumors as normal values. Previously, we had already observed normalization of hematopoiesis when treating mice with the Tdp1 inhibitor OL7-43 [53], which is also a derivative of usnic acid, but with modifications different from inhibitor **7**. The mechanism of the effect of Tdp1 inhibitors on hematopoiesis is unknown and requires further study.

## 3. Materials and Methods

### 3.1. Chemistry

The analytical and spectral studies were conducted at the Chemical Service Center for the collective use of Siberian Branch of the Russian Academy of Science. The 1H and ^13^C-NMR spectra for solutions of the compounds in CDCl_3_ were recorded on a Bruker AV-400 spectrometer (Bruker Corporation, Hanau, Germany; operating frequencies 400.13 MHz for ^1^H and 100.61 for ^13^C). The residual signals of the solvent were used as references (δH 7.27, δC 77.1 for CDCl_3_). Merck silica gel (63–200 μ) was used for the column chromatography. Thin-layer chromatography was performed on TLC Silica gel 60F_254_ (Merck KGaA, Darmstadt, Germany). The target compounds reported in this manuscript had a purity of at least 95% (HPLC). All chemicals were used as described unless otherwise noted. Reagent-grade solvents were redistilled prior to use. Synthetic starting materials, reagents, and solvents were purchased from Sigma-Aldrich (St. Louis, MO, USA), Acros Organics (Geel, Belgium), and AlfaAesar (Heysham, UK).

(R)-(+)-Usnic acid (+)**-9** was purchased from Zhejiang Yixin Pharmaceutical Co., Ltd. (Lanxi, China). Compounds **3**,**4** were obtained using a known procedure. The spectra of the substances coincided with that in the literature [31,32].

#### Synthesis of Hybrid Compounds **5**–**8**

The hydrazono thiazole **3** or **4** (1 mmol) was dissolved in 35 mL of ethanol. The corresponding amine (3 mmol) was added. The resulting solution was stirring with reflux for 15–20 h (TLC monitoring). After that, the solution was diluted with water. The resulting precipitate was filtered off, washed with water, and air-dried. The target compound was isolated after column chromatography (eluent: dichloromethane-hexane-triethylamine 50:50:1).

(4E)-10-{2-[(E)-2-[(5-bromothiophenyl)methylidene]hydrazin-1-yl]-1,3-thiazole-4-yl}-4-(1-{[3-(3,5-di-tret-butyl-4-hydroxyhenyl)propyl]amino}ethylidene)-11,13-dihydroxy-2,12-dimethyl-8-oxatricyclo [7.4.0.02,7]trideca-1(13),6,9,11-tetraen-3,5-dione **5**.



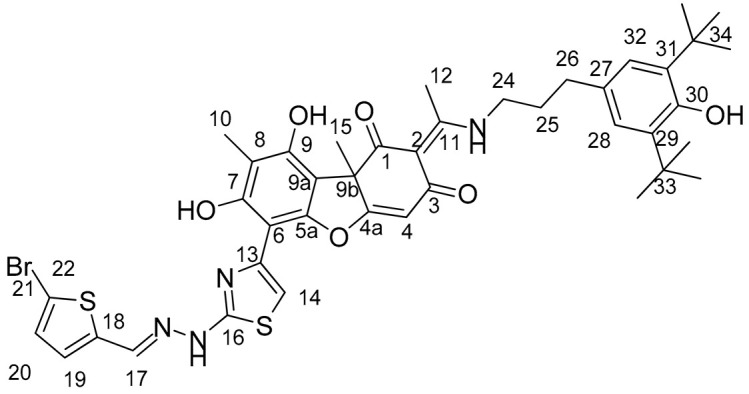



Orange amorphous powder. Yield 27%. T_decomp_. = 136–138 °C. _1_H NMR (CDCl_3_, δ): 1.69 (3H, s, H-15), 2.11 (3H, s, H-10), 2.54 (3H, s, H-12), 5.83 (1H, s, H-4), 6.55 (2H, d, J = 8.1 Hz, H-24), 6.79 (1H, s, H-19), 6.89 (1H, s, H-20) 7.14 (1H, c, H-14), 7.54 (2H, d, J = 8.1 Hz, H-25), 7.71 (1H, s, H-17), 10.87 (1H, s, OH-9), 15.08 (1H, s, OH-3). 13C NMR (CDCl3, δ): 8.37 (C-10), 20.32 (C-12), 31.77 (C-15), 57.83 (C-9b), 97.23 (C-9a) 101.27 (C-14), 102.93 (C-6), 104.45 (C-8), 104.52 (C-4), 107.81 (C-2), 121.61 (C-27), 123.55 (C-21), 127.19 (C-25, C-29), 127.89 (C-19, C-23), 131.61 (C-20, C-22), 132.47 (C-18), 132,63 (C-26, C-28), 135.16 (C-24), 140.89 (C-17), 151.54 (C-9), 151.85 (C-7), 155.92 (C-5a), 166.07 (C-16), 173.15 (C-11), 175.73 (C-4a), 191.12 (C-3), 199.04 (C-1). IR (cm-1): 466.71, 503.35, 534.21, 567.00, 667.28, 696.21, 752.14, 769.49, 792.64, 840.85, 971.99, 1056.85, 1074.21, 1110.85, 1170.63, 1187.99, 1207.28, 1234.28, 1278.63, 1305.63, 1357.70, 1432.92, 1463.77, 1515.84, 1556.34, 1592.99, 1618.06, 1695.20, 2852.33, 2923.69, 2956.48, 3440.54.

(4E)-4-{1-[(5-bromothiophenyl)amino]ethilidene}-10-{2-[(E)-2-[(4-bromophenyl) methylidene]hydrazin-1-yl]-1,3-thiazol-4-yl}-11,13-dihydroxy-2,12-dimethyl-8-oxatricyclo [7.4.0.02,7]trideca-1(13),6,9,11-tetraen-3,5-dione **6**.



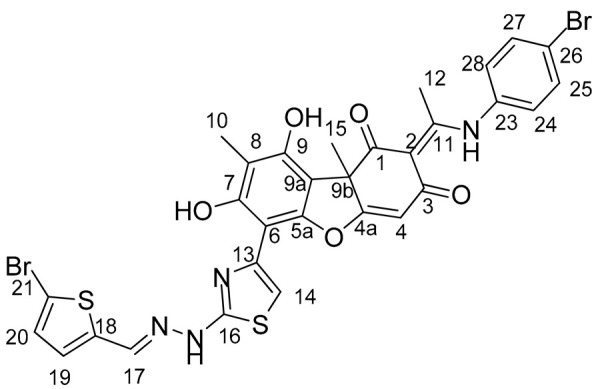



Dark-brown amorphous powder. Yield 70%. T_decomp_. = 135–137 °C. ^1^H NMR (CDCl_3_, δ): 1.69 (3H, s, H-15), 2.11 (3H, s, H-10), 2.54 (3H, s, H-12), 5.83 (1H, c H-4), 6.55 (2H, d, J = 8.1 Hz, H-24), 6.79 (1H, s, H-19), 6.89 (1H, s, H-20) 7.14 (1H, s, H-14), 7.54 (2H, d, J = 8.1 Hz, H-25), 7.71 (1H, s, H-17), 10.87 (1H, s, OH-9), 15.08 (1H, s, OH-3). 13C NMR (CDCl3, δ): 8.37 (C-10), 20.32 (C-12), 31.77 (C-15), 57.83 (C-9b), 97.23 (C-9a) 101.27 (C-14), 102.93 (C-6), 104.45 (C-8), 104.52 (C-4), 107.81 (C-2), 121.61 (C-27), 123.55 (C-21), 127.19 (C-25, C-29), 127.89 (C-19, C-23), 131.61 (C-20, C-22), 132.47 (C-18), 132,63 (C-26, C-28), 135.16 (C-24), 140.89 (C-17), 151.54 (C-9), 151.85 (C-7), 155.92 (C-5a), 166.07 (C-16), 173.15 (C-11), 175.73 (C-4a), 191.12 (C-3), 199.04 (C-1). IR (cm-1): 501.42, 545.78, 688.49, 723.21, 740.57, 784.92, 811.92, 844.71, 927.64, 948.85, 970.06, 1010.56, 1031.78, 1060.71, 1091.56, 1114.71, 1170.63, 1205.35, 1278.63, 1303.70, 1346.13, 1369.27, 1398.20, 1421.35, 1459.92, 1488.84, 1544.77, 1594.92, 1621.92, 1693.27, 2923.69, 2973.83, 3089.55, 3440.54.

(4E)-10-{2-[(E)-2-[(4-bromophenyl)methylidene]hydrazin-1-yl]-1,3-thiazole-4-yl}-4-(1-{[3-(3,5-di-tret-butyl-4-hydroxyhenyl)propyl]amino}ethylidene)-11,13-dihydroxy-2,12-dimethyl-8-oxatricyclo [7.4.0.02,7]trideca-1(13),6,9,11-tetraen-3,5-dione **7**.



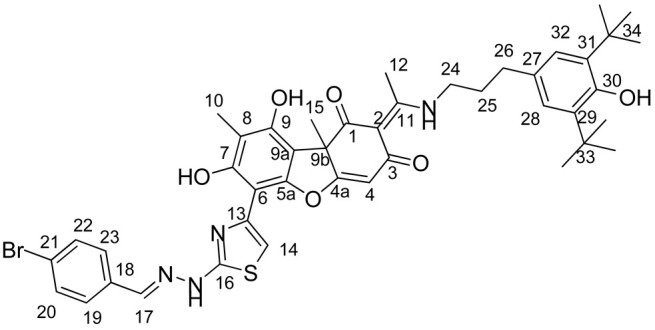



Orange amorphous powder. Yield 34%. T_decomp_. = 136–139 °C. ^1^H NMR (CDCl3, δ): 1.69 (3H, s, H-15), 2.11 (3H, s, H-10), 2.54 (3H, s, H-12), 5.83 (1H, s, H-4), 6.55 (2H, d, J = 8.1 Hz, H-24), 6.79 (1H, s, H-19), 6.89 (1H, s, H-20) 7.14 (1H, s, H-14), 7.54 (2H, d, J = 8.1 Hz, H-25), 7.71 (1H, s, H-17), 10.87 (1H, s, OH-9), 15.08 (1H, s, OH-3). 13C NMR (CDCl3, δ): 8.37 (C-10), 20.32 (C-12), 31.77 (C-15), 57.83 (C-9b), 97.23 (C-9a) 101.27 (C-14), 102.93 (C-6), 104.45 (C-8), 104.52 (C-4), 107.81 (C-2), 121.61 (C-27), 123.55 (C-21), 127.19 (C-25, C-29), 127.89 (C-19, C-23), 131.61 (C-20, C-22), 132.47 (C-18), 132,63 (C-26, C-28), 135.16 (C-24), 140.89 (C-17), 151.54 (C-9), 151.85 (C-7), 155.92 (C-5a), 166.07 (C-16), 173.15 (C-11), 175.73 (C-4a), 191.12 (C-3), 199.04 (C-1). IR (cm-1): 511.07, 534.21, 688.49, 736.71, 769.49, 819.64, 846.64, 875.56, 927.64, 950.78, 1008.63, 1049.13, 1068.42, 1089.63, 1116.63, 1170.63, 1213.06, 1232.35, 1278.63, 1303.70, 1342.27, 1367.35, 1394.35, 1436.77, 1465.70, 1558.27, 1591.06, 1616.13, 1697.13, 2869.69, 2923.69, 2954.55, 3070.26, 3396.19, 3633.40.

(4E)-4-{1-[(4-bromophenyl)amino]ethilidene}-10-{2-[(E)-2-[(4-bromophenyl)methyl idene]hydrazin-1-yl]-1,3-thiazol-4-yl}-11,13-dihydroxy-2,12-dimethyl-8-oxatricyclo [7.4.0.02,7]trideca-1(13),6,9,11-tetraen-3,5-dione **8**.



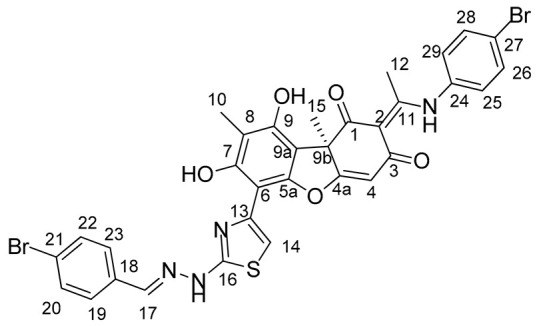



Dark-brown amorphous powder. Yield 75%. T_decomp_. = 135–137 °C. ^1^H NMR (CDCl3, δ): 1.65 (3H, s, H-15), 2.18 (3H, s, H-10), 2.55 (3H, s, H-12), 5.78 (1H, c H-4), 7.04 (2H, д, J = 7.6 Hz, H-25), 7.12 (1H, c, H-14), 7.25 (1H, s, H-17), 7.28 (2H, д, J = 8.8 Hz, H-19), 7.36 (2H, д, J = 8.8 Hz, H-20), 7.54 (2H, д, J = 7.6 Hz, H-26), 9.33 (1H, s, NH), 10.93 (1H, s, OH-9), 15.08 (1H, s, OH-3). 13C NMR (CDCl3, δ): 8.37 (C-10), 20.32 (C-12), 31.77 (C-15), 57.83 (C-9b), 97.23 (C-9a) 101.27 (C-14), 102.93 (C-6), 104.45 (C-8), 104.52 (C-4), 107.81 (C-2), 121.61 (C-27), 123.55 (C-21), 127.19 (C-25, C-29), 127.89 (C-19, C-23), 131.61 (C-20, C-22), 132.47 (C-18), 132,63 (C-26, C-28), 135.16 (C-24), 140.89 (C-17), 151.54 (C-9), 151.85 (C-7), 155.92 (C-5a), 166.07 (C-16), 173.15 (C-11), 175.73 (C-4a), 191.12 (C-3), 199.04 (C-1). IR (cm-1): 507.21, 543.85, 649.92, 690.42, 763.71, 806.14, 819.64, 842.78, 873.64, 923.78, 948.85, 1010.56, 1031.78, 1060.71, 1118.56, 1170.63, 1193.78, 1276.70, 1303.70, 1346.13, 1369.27, 1396.27, 1419.42, 1459.92, 1486.92, 1542.84, 1591.06, 1621.92, 1693.27, 2759.76, 2923.69, 2975.76, 3087.62, 3440.54.

### 3.2. Biological Assays

#### 3.2.1. Tdp1 Activity

An oligonucleotide 5′-[FAM] AAC GTC AGGGTC TTC C [BHQ1]-3′ synthesized in the laboratory Laboratory of Nucleic Acids Chemistry at the Institute of Chemical Biology and Fundamental Medicine (Novosibirsk, Russia) was used as a biosensor. Removal of the quencher (BHQ1) due to enzyme activity leads to increased fluorescence of FAM. The reaction was carried out at different concentrations of inhibitors (1.5% of DMSO, Sigma, St. Louis, MO, USA, in the control samples). The reaction mixtures contained Tdp1 buffer (50 mM Tris-HCl pH 8.0, 50 mM NaCl, and 7 mM β-mercaptoethanol), 50 nM biosensor, and an inhibitor being tested. Purified Tdp1 (1.5 nM) triggered the reaction.

The fluorescence was measured using POLARstar OPTIMA fluorimeter (BMG LABTECH, GmbH, Ortenberg, Germany). The values of IC_50_ were determined in minimum three independent experiments and were calculated using embedded software MARS Data Analysis 2.0 (BMG LABTECH, GmbH, Ortenberg, Germany).

#### 3.2.2. Cytotoxicity Assays

Cytotoxicity of the compounds against human tumor cell lines: HeLa (cervical cancer), HCT-116 (colorectal cancer), A-549 (lung cancer), MCF-7 (breast cancer), T98G (glioblastoma), HepG2 (hepatocarcinoma), and nontumor cell lines: MRC-5 (lung fibroblasts) and HEK293A (human embryonic kidney) was examined using the MTT test [54]. HCT-116 and MRC-5 cell lines were provided by the Cell Culture Bank of the State Research Center for Virology and Biotechnology Vector, Novosibirsk, Russia. A-549, MCF-7, and T98G cell lines were obtained from the Russian Cell Culture Collection (RCCC) Institute of Cytology RAS (St. Petersburg, Russia). HEK293A cells were purchased (ThermoFisher Scientific, Waltham, MA, USA).

The cell culture medium DMEM medium (Invitrogen, Carlsbad, CA, USA) contained 10% fetal bovine serum (FBS) (Invitrogen), penicillin (100 units/mL), and streptomycin (100 µg/mL). Cells were seeded into 96-well plates at 5000 cells per well and grown at 37 °C and 5% CO_2_ in a humid atmosphere. At 30–50% confluence, the tested compounds were added to the medium to a final DMSO content of 1%. Control wells contained 1% DMSO. To determine the cytotoxicity of Tdp1 inhibitors, the cells were treated with compounds with concentrations ranging from 1 to 100 µM for 72 h at 37 °C for 72 h. All measurements were repeated a minimum of twice.

To study the effect of the compounds on the cytotoxicity of topotecan, we used different concentrations of aqueous topotecan against a background of 10 μM Tdp1 inhibitors. Cells treated with compounds alone without topotecan were used as controls.

#### 3.2.3. Laboratory Animals and Tumor Models

Female C57BL mice with a body weight of ~19–21 g were used in the study. Mice were housed in plastic cages with a wood chip bedding and had free access to water and food. Lewis lung carcinoma (LLC) and Krebs-2 ascites were used as experimental tumor models. The transplantable tumors were obtained from the cell bank of the Institute of Cytology and Genetics (Novosibirsk, Russia). The LLC mouse model is the most common model of lung cancer; LLC cells remain tumorigenic and capable of metastasis into the lung in C57BL mice. Tumor tissue was minced and resuspended in 0.9% NaCl prior to grafting. LLC was injected into the right thigh of mice at 800,000 cells in 0.2 mL per mouse.

Strain-nonspecific Krebs-2 ascitic carcinoma was grafted to C57BL mice. Tumor cells were suspended in 0.9% NaCl prior to grafting and injected intraperitoneally (2 million cells per mouse, in 0.2 mL). An ascitic tumor develops after intraperitoneal inoculation, has low immunogenicity, and produces no metastasis during the experiment at the used dose of tumor cells.

All experiments with mice were carried out in accordance with the protocols approved by the Inter-Institute Commission on Bioethics of the Institute of Cytology and Genetics of the Siberian Branch of the Russian Academy of Sciences No. 21.11 dated May 30, 2014, and directive 2010/63/EU.

#### 3.2.4. Antitumor and Antimetastatic Effects of Compound **7** Used as Monotherapy or Combined with Topotecan in the LLC Model

Previously, we divided the animals into 6 groups of 6–7 individuals in each, the treatment was carried out once 7 days after tumor transplantation:-Group 1—Intact control—mice of this group were inoculated with a tumor, receiving 0.2 mL of saline intraperitoneally;-Group 2—DMSO-Tween-80—mice were injected once intragastrically with a probe 200 μL of a solution containing saline, Tween-80 (10%) and DMSO (15%), a solvent for **7**;-Group 3—Tpc (aqueous solution)—mice were injected once intraperitoneally with 200 µL of Tpc at a single dose of 1 mg/kg;-Group 4—Tpc + **7** i/g—mice received Tpc once as described for group 3 and **7** in the form of 200 µL of suspension (Tween-80 (10%) and DMSO (15%) in saline) once intragastrically at a dose of 50 mg/kg;-Group 5—**7** i/g—mice were injected with **7** as described for group 4;-Group 6—Tpc + **7** i/p—mice were injected with topotecan once as described for groups 3 and **7** in the form of 200 µL of suspension (Tween-80 (10%) and DMSO (15%) in saline) once intraperitoneally at a dose of 50 mg/kg;-Group 7—**7** i/p—mice were injected with **7** as for group 6.

The substances were administered as a single injection in a volume of 0.2 mL on day 13 after tumor grafting. The anticancer effect was evaluated by the primary tumor growth and the number of metastases in the lungs of all mice. Metastases were counted after fixation in 10% formalin under an MBI-3 microscope (LOMO, Saint Petersburg, Soviet Union) at a threefold magnification.

#### 3.2.5. Antitumor Effect of Compound **7** Used as Monotherapy or in Combination with Topotecan in the Krebs-2 Ascitic Carcinoma Model

The mice were also divided into groups of 6–7 individuals. Treatment was carried out once on the 4th day after tumor transplantation:-Group 1—Intact control—mice of this group were inoculated with a tumor, receiving 0.2 mL of saline intraperitoneally;-Group 2—DMSO-Tween-80—mice were injected once intragastrically with a probe 200 μL of a solution containing saline, Tween-80 (10%), and DMSO (15%);-Group 3—DMSO-Tween-80—mice were injected once intraperitoneally with a solvent for **7**;-Group 4—Tpc (aqueous solution)—mice were injected once intraperitoneally with 200 µL of Tpc at a single dose of 1 mg/kg;-Group 5—Tpc + **7** i/g—mice received Tpc once as described for groups 3 and **7** in the form of 200 µL of suspension in Tween-80-DMSO once intragastrically at a dose of 50 mg/kg;-Group 6—Tpc + **7** i/p—mice were injected with topotecan once as described for groups 3 and **7** in the form of 200 µL of suspension Tween-80-DMSO once intraperitoneally at a dose of 50 mg/kg;-Group 7—**7** i/g—mice were injected with **7** as described for group 5;-Group 8—**7** i/p—mice were injected with **7** as described for group 6.

The ascitic tumor weight and the cancer cell concentration in the ascitic fluid (as counted with a Goryaev chamber) were evaluated at the end of the experiment to estimate the effects of the substances. To prepare the samples for cell counting with a Goryaev chamber, 10 μL of the ascitic fluid was combined with 400 μL of saline (a 40-fold dilution). Cells were counted in five large squares, each divided into 16 small squares (80 squares in total) at a low magnification.

#### 3.2.6. The Toxic Effect of Drugs and Their Combination

The toxic effects of the substances were inferred from the changes in body weight during the experiment (the mice with LLC were weighed on days 1, 3, 5, 7, 10, 12, 14, 16, and 18 after grafting) and the liver and spleen weight indices at the end of the experiment.

#### 3.2.7. The Effects of the Substances on the Differential Blood Count

White and red blood cell counts were determined using a Goryaev chamber using a BIOSCOP-1 microscope (LOMO-MA, St. Petersburg, Russia).

#### 3.2.8. Statistical Analysis

Measurements and calculations of primary data in in vivo experiments were carried out in a blinded manner. Statistical processing of data was carried out using one-way ANOVA (STATISTICA software version 12.5). Post hoc assessment was performed using Tukey’s honestly significant difference (HSD) test. Data with *p* < 0.05 were considered statistically significant.

Pairwise comparison of cell survival at different concentrations of topotecan in the presence and absence of Tdp1 inhibitors was performed using the Mann–Whitney test.

## 4. Conclusions

In conclusion, we have designed and synthesized a class of Tdp1 inhibitors based on usnic acid, which are hybrid molecules carrying two modifications in rings A and C, leading to effective inhibition of the enzyme and reduced toxicity of the compounds. The compounds inhibit Tdp1 in the submicromolar and micromolar concentration range, and compounds with a di-tert-butyl group in ring C are an order of magnitude more effective inhibitors than compounds with a bromophenol group. Biderivatized compounds are nontoxic or mildly toxic to a wide panel of cell lines representing various tumor models, as well as cells of non-cancerous origin. Some new derivatives sensitize the cytotoxic effect of topotecan in vitro. Lead compound **7** enhances the antitumor and antimetastatic effect of topotecan in vivo. We also showed that this compound does not increase organ (liver and spleen) indices and does not reduce the body weight of mice, which indicates that accompanying therapy with compound **7** could be well tolerated. In addition, compound **7** in combination with topotecan reduced the number of leukocytes in mice with Krebs-2 carcinoma and increased the number of red blood cells to normal counts. All effects of the topotecan-**7** combination are more pronounced when last administered intraperitoneally. Thus, we have obtained a promising combination of drugs for antitumor therapy.

## Data Availability

The data presented in this study are available in Appendix A.

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
