# Peer review of "Enhancement of the Antitumor and Antimetastatic Effect of Topotecan and Normalization of Blood Counts in Mice with Lewis Carcinoma by Tdp1 Inhibitors—New Usnic Acid Derivatives"

_ijms, 2024, doi:10.3390/ijms25021210_

Round 1

Reviewer 1 Report

Comments and Suggestions for Authors

In the present work Kornienko et al. reported on the role of Tdp1 inhibitors as enhancement and anti-metastatic regulators with respect to topotecan anti-tumor therapy. Their work is very interesting and I have enjoyed reading the present manuscript. It is well-written and the topic is interesting. The present work has merit for publication. 

The concept presented in this manuscript is very interesting, especially as the authors mention that Tdp1 is a key-molecule in chemoresistance acquisition it is of outmost importance. The authors should elaborate a little bit more on that. For example, expand their report on drugs used as combinatorial therapy with Tdp1 inhibitors. The authors should report the significance levels in all tables and figures, in order to show if the observed differences were of significance. This applies to both the manuscript values as well as the supplementary data presented. Although in some cases is obvious, still the authors should present the significance levels.

In addition, it appeared that the vehicle manifested significant anti-tumor toxicity in in vivo experiments. Please comment on this. How is to explain that vehicle had significant (if it is so, since p-values are missing) anti-tumor, anti-metastatic activities (for example, in figure 3, vehicle manifests comparable effect to the other compounds). Further on, in the in vitro experiments the vehicle experiment is missing. The authors should present the in vitro toxicity of the vehicle/solute in cells in culture. 

Overall, the authors have done a great deal of effort and experimentation, which is to their appraisal and furthermore, they have done an excellent work presenting their hypothesis.

Comments on the Quality of English Language

None

Author Response

In the present work Kornienko et al. reported on the role of Tdp1 inhibitors as enhancement and anti-metastatic regulators with respect to topotecan anti-tumor therapy. Their work is very interesting and I have enjoyed reading the present manuscript. It is well-written and the topic is interesting. The present work has merit for publication.

Response: Thanks a lot to Reviewer for the positive assessment of our work and useful suggestions and comments. We tried to take them into account, which really makes the article better and easier to understand.

The concept presented in this manuscript is very interesting, especially as the authors mention that Tdp1 is a key-molecule in chemoresistance acquisition it is of outmost importance.

The authors should elaborate a little bit more on that. For example, expand their report on drugs used as combinatorial therapy with Tdp1 inhibitors.

Response: We realized that we were carried away by describing the role of Tdp1 in the resistance of tumors to chemotherapy, as well as its inhibitors, and said nothing at all about Top1 inhibitors, the effect of which Tdp1 inhibitors are designed to enhance. We thank Reviewer for the helpful comments and add to the Introduction a few sentences about known Top1 inhibitor chemotherapy drugs, such as topotecan, and their relationship with Tdp1. Unfortunately, Tdp1 inhibitors have not yet been included in clinical trials and are not used to treat humans. In fact, only our team has gotten to the point of using combinations of Tdp1 inhibitors with known chemotherapy drugs in animals, and only combinations with topotecan are successful. We tried a Tdp1 inhibitor in combination with the PARP1 inhibitor olaparib, but did not see a synergistic effect (Kornienko et al., Mol. Biol. 2023, 57, 214–224. DOI: 10.1134/S0026893323020127). It has also been suggested, based on in vitro data, that Tdp1 inhibitors may potentiate the effect of other chemotherapy drugs (see review Cameaux&VanWaardenburg, Drug Metab. Rev. 2014, 46, 494–507. https://doi.org/10.3109/03602532.2014.971957), but these works have not yet received further development.

The authors should report the significance levels in all tables and figures, in order to show if the observed differences were of significance. This applies to both the manuscript values as well as the supplementary data presented. Although in some cases is obvious, still the authors should present the significance levels.

Response: Thank you. We have made the necessary changes. In Table 2, instead of comments, we presented numerical values and noted IC50 values with significantly different graphs at two (if there are 4 points) or three (if there are 5 points) midpoints. The significance of differences was compared in pairs with the Mann-Whitney test. The first points are 100% for all graphs. We did not consider the last points with the highest concentration of topotecan, since in some cases there were too few living cells left. The mouse experiments’ graphs show significant differences according to Tukey's test.

In addition, it appeared that the vehicle manifested significant anti-tumor toxicity in in vivo experiments. Please comment on this. How is to explain that vehicle had significant (if it is so, since p-values are missing) anti-tumor, anti-metastatic activities (for example, in figure 3, vehicle manifests comparable effect to the other compounds).

Response: The intrinsic antitumor properties of DMSO have been reported in the literature (Goto et al., BrJCancer, 1996, 74, 546-554; Wang et al, Plos One, 2012, 7, e33772. doi: 10.1371/journal.pone.0033772), although other researchers did not confirm it (Elisia et al., Plos One, 2016, 11, e0152538. doi: 10.1371/journal.pone.0152538). In our case, the vehicle effect looks impressive, but according to Tukey’s test, the difference between the intact control groups (mice received saline solution) and the vehicle group was not significant. The range of the number of metastases in the control group that received saline solution ranges from 2 to 30. In the group that received DMSO, the range of values is from 7 to 12, that is, it overlaps 100% with the first group. A similar picture is observed with the primary node, the values in the two groups overlap (from 3.7 to 4.9 for control and from 3.2 to 4.6 for DMSO). We added primary data and the results of their processing according to the Tukey criterion to the supplementary (file “Mice primary data.xlsx”).

Further on, in the in vitro experiments the vehicle experiment is missing. The authors should present the in vitro toxicity of the vehicle/solute in cells in culture.

Response: In in vitro experiments, 1% DMSO was present in control wells, it was specified in section 4.2.2. In the wells with Tdp1 inhibitors, the same concentration of DMSO was used. A mixture of 15% DMSO and 10% Tween 80 (a vehicle in in vivo experiments) cannot be tested in vitro on cell cultures because it is lethal to the cell lines.

Overall, the authors have done a great deal of effort and experimentation, which is to their appraisal and furthermore, they have done an excellent work presenting their hypothesis.

 Response: The authors sincerely thank the Reviewer for his kind words. Such a high assessment of our work is very pleasant and inspires us for further work. 

Reviewer 2 Report

Comments and Suggestions for Authors

I found the study interesting and promising.  Generation and very preliminary characterization of new putative inhibitors of Tdp1 to overcome TopI inhibitors resistance.

These revisions must be done in my opinion before publishing:

Major points:

1. give more details on the chemical synthesis, justify why the yield is very low/variable for some compounds, present data on purity and solubility and pharmacology (eg in vivo metabolism of the drug);

2. evaluate the effect of compound 7 on Cdp1 KO/KD cells;

3. substitute table 2 with CC50 data and scientific relevant data;

4. the inspection of metastasis is not supported by IHC data or markers for tumor diffusion;

5. compound 7 must be better characterized adding pharmacology data;

6. orthogonal methods are required to sustain in vitro inhibition of Tdp1;

7. in vivo inhibition of Tdp1 must be tested;

8. table 2 must contain values, not general comments; data about in vivo experiments must be supported by IHC or other approaches to sustain conclusion; not clear the treatments groups; not clear which treatment in the control group.

Minor points:

It is not clear if tested cells are resistant to topotecan (acquired resistance or innate resistance?)

Comments on the Quality of English Language

Only minor changes are required

Author Response

The authors thank the Reviewer for his attention to our work and useful comments. We tried to answer them all. Please find the attached file with the answers.

Round 2

Reviewer 2 Report

Comments and Suggestions for Authors I welcome the changes made by the authors. However, two aspects still require clarification and further evaluation. First, the data submitted in response to the reviewer and obtained in HEK293 illustrate the lack of synergistic effect resulting from treatment with topotecan and compound 7 (referred to as compound 185) in both WT and Tdp1 KO. These data need to be included as stated in the paper (in the main body or supplementary) and discussed (although the model is not the best as I would have preferred an evaluation on tumor cell lines). Why is this the case? These data do not sustain the initial hypothesis, change the model for the precise evaluation. Secondly, I would also like to see a hematoxylin/eosin of lung metastases and their numbers. Both evaluation methods were introduced almost 30 years ago (https://doi.org/10.1016/0092-8674(94)90200-3 Comments on the Quality of English Language

English quality is good

Author Response

Iwelcomethechangesmadebytheauthors.However,twoaspectsstillrequireclarificationandfurther evaluation.

Response: We thank the Reviewer for the  kind criticism of our work and useful comments.

First,thedatasubmittedinresponsetothereviewerandobtainedinHEK293illustratethelackofsynergisticeffectresultingfromtreatmentwithtopotecanandcompound7(referredtoascompound185)in bothWTandTdp1KO.Thesedataneedtobeincludedasstatedinthepaper(inthe mainbody orsupplementary)anddiscussed(althoughthemodelisnotthebestasIwouldhavepreferredanevaluationontumorcelllines).Whyis thisthecase? These data do not sustain the initial hypothesis, change the model for the precise evaluation.

Response: Thank you, we have included this data to the supplementary. We are currently working on a creation of the tumor cell line with Tdp1 knockout and plan to test the effect of inhibitors of this enzyme on the cytotoxic effect of topotecan on this knockout cell line. Selecting a cancer line and creating a knockout model is a separate task for this time for us. Of course, if we are successful, the Reviewer is right, testing on a cancer model will be necessary for our future work in more detailed studies of the action of the leader compound.

Secondly, I would also like to see a hematoxylin/eosin of lung metastases and their numbers. Both evaluation methods were introduced almost 30 years ago (https://doi.org/10.1016/0092-8674(94)90200-3

Response: Unfortunately, due to an oversight, we did not save the lung samples after the experiment. Thus it is not possible to make hematoxylin/eosin for this paper. Sure now we sorry about it. But we plan to continue our research on working with compound 7 also studying the pharmacokinetics and adjusting the administration regimen, and we will definitely stain the lungs with metastases. Please find in the attached file an example of lungs with metastases

Round 3

Reviewer 2 Report

Comments and Suggestions for Authors

It is a pity to do not have the HE stainig for lung metastasis as well as to do not have the Tdp1 KO in cancer cells

Comments on the Quality of English Language

Good quality